# Wear and Corrosion Behavior of Cold-Sprayed Cu-10Sn Coatings

Ismail Ozdemir [1,*], Bahattin Bulbul [2], Thomas Grund [1] and Thomas Lampke [1]

1   Materials and Surface Engineering Group, Chemnitz University of Technology, Erfenschlager Str. 73, 09125 Chemnitz, Germany
2   Eksel Bimetal Sintering and Casting Factory, 35100 Izmir, Turkey; bahattin@ekselbimetal.com.tr
*   Correspondence: ismail.oezdemir@mb.tu-chemnitz.de

**Abstract:** Gas-atomized Cu-10Sn powders as a potential substitute for sintered bronze layers are usually employed in plain-bearing shells produced by cold spray (CS) processes on steel substrates (AISI 1010). In this study, the effective thickness, i.e., approx. 450 μm, of the bronze overlay required for the bearing shell was successfully and cost-effectively deposited in a short time. A ball-on-disc test setup was used to explore the tribological behavior of cold-sprayed bronze coatings under dry sliding conditions, and the electrochemical corrosion behaviors of sprayed coatings at room temperature were also evaluated by using the potentiodynamic scanning (PDS) technique in acidic (0.01 M Na$_2$SO$_4$) and alkaline (3.5% NaCl) environments. The characterization of the sprayed bronze coatings revealed no formation of oxidation or new phases during cold spraying and that the coatings were well-adhered to the substrates, implying good bonding. The wear results demonstrated that, as the load and sliding distance increased, the friction coefficients and wear rates of the sprayed coatings decreased. The PDS results showed that the corrosion resistance of the Cu-10Sn coating layer in an acidic environment is higher than that in an alkaline environment. In addition, the coated layer presented no passivation or pitting onset due to the heavy corrosion reaction in an alkaline solution.

**Keywords:** plain bearing; cold spray; Cu-10Sn coating; corrosion; wear





## 1. Introduction

Plain bearings in automobile engines provide relative motion between the crankshaft, engine block, and connecting rods and are subjected to varying loads through the hydrodynamic oil layer. Roller, plain, air, and magnetic bearings are examples of several types of bearings that can be found in half shells or bushes [1,2]. This study focuses on plain bearings mainly composed of bimetals and trimetals for combustion engines. These bearings are necessary to separate moving surfaces, which is accomplished by the application of lubricating hydrodynamic oil. Such bearings should provide sufficient fatigue wear, corrosion resistance, compatibility, conformability, and embeddability during shaft revolutions via hydrodynamic action.

On the other hand, increasing demands on combustion engines, such as downsizing and decreasing gas emissions, require the development of novel bearing materials that provide the longest feasible lifetimes at greater operating pressures. To achieve current engine performance requirements, plain-bearing layers with tailored properties should be layered on steel shells [1].

Copper-based lining layers are commonly utilized as bearing materials since they possess good thermal characteristics as well as excellent wear and corrosion resistance [3–6]. Due to the natural embeddability and antifriction characteristics of lead, lead-based bearing alloys were extensively used to prevent bearing surface deterioration and subsequent engine failure during running [7]. The restriction of lead use owing to environmental and health concerns has resulted in a greater interest in developing lead-free bearing materials

for a variety of applications [8]. Unlu and Atik investigated the tribological characteristics of CuSn10 and CuZn30 cast alloys and found that these alloys had the greatest wear rates when compared to pure Cu, Zn, and Sn as bearing materials [3]. The results showed that using cast Cu alloyed bearings was not a viable option for reaching the greater load-carrying capability in long-term use.

Bearing materials that are widely manufactured via the sintering technique were assumed to have sufficient friction, embeddability, and conformability properties. To compensate for these limitations, additional coatings called overlays, such as Sn-based thin films, plated on bearing surfaces outperformed Pb-based overlays in terms of tribological performance [4]. Instead of addressing metallic or polymeric overlays on the bearing surface, the study focused on depositing tribolayers (at least 0.25 mm thick) with improved wear resistance and corrosion resistance. Chemical interactions between lubricating oil particles and bearing surfaces are well-known to cause material loss during engine operation. It was recently demonstrated that Cu-based wear-resistant coatings sprayed by cold spraying over components subjected to varying operating conditions could fulfil service-performance criteria [9]. The primary advantage of the cold spray method is its capacity to create very thick layers on a wide range of substrates with strong bond strength due to residual stress induced in the vicinity of the interface, which enables metallurgical bonding [10]. Additionally, as CS is such a versatile technique, it allows for the direct deposition of even the most complex geometric patterns, resulting in the development of innovative bearing components. As compared to conventional solid-state bearing production methods, this provides cost reductions as well as improved efficiency.

Low-pressure cold spraying (LPCS) has recently been recognized as a viable method for depositing bearing materials with wear performance at least similar to that of traditional bearing surfaces [11]. Likewise, thermally sprayed bronze coatings with solid lubricants for highly stressed sliding bearings revealed much lower material loss when exposed to wear tests [12]. Moreover, to improve the self-lubricating effect of the bearing surface and thus increase the life cycle of the bearing material, a CuSn10 layer with an h-BN/graphite coating was deposited on engine-bearing steel surfaces using cold and liquid spraying methods [13]. The low friction coefficient of the cold-sprayed coating examined under dry sliding circumstances was linked to the formation of the self-lubricating effect of the h-Bn/graphite coating. Nevertheless, it was demonstrated that the addition of a solid lubricant to cold-sprayed Cu coatings might impair the interface bonding strength, leading to wear-related failure of the component in service [9]. As a matter of fact, the hard and soft properties of bearing materials must be optimized in order to resist wear and enable impurities embedded at the surface to prevent excessive wear. Cold-sprayed coatings, in general, have superior corrosion characteristics due to their high densities, phase purities, and homogeneous microstructures. Additionally, the study revealed that spraying copper with hard ceramic particles, which allows for peening effects during deposition and therefore porosity reduction, could significantly increase the corrosion resistance of cold-sprayed copper coatings [14], corrosion resistance of Cu Cs. Similarly, post-treatments, such as ultrasonic shot peening, might improve the corrosion resistance of cold-sprayed copper due to reduced porosity and higher bonding strength [15] Cu/Ni corrosion CS 2020. It was also previously found that hard reinforcing particles increased the wear resistance of CS copper coatings, while having no effect on corrosion resistance when tested in a Cl- environment, where oxygen and chlorine were concentrated at the pore and particle matrix interface [16]. It was also claimed that by using suitable spray parameters, powders, and post-treatments, a virtually fully dense structure in a CS cu coating against corrosion failure, notably for anodic protection, could be obtained. Besides that, corrosion damages, in addition to corrosion resistance, could be refurbished with strong corrosion-resistant materials using cold spraying, resulting in increased component lifespan [17].

Thus, this study focused on the tribological and corrosion behavior of Cu-10Sn coatings deposited onto typical steel surfaces by the LPCS method. The pile-up of CS layers and their

adhesion to the substrate were evaluated by means of cross-sectional and XRD analysis. The corrosion behavior of the deposited coatings was investigated at room temperature in both acidic and alkaline solutions by means of PDS.

## 2. Experimental Procedure

### 2.1. Powder Feedstock, Coating Deposition, and Characterization

Before coating deposition, flat-form steel substrates $40 \times 40 \times 8$ mm in dimensions were cut from conventional steel shells (AISI 1010) using an abrasive cutting machine and were grit-blasted with corundum abrasive particles prior to coating deposition. To increase the adhesive strength of the steel substrates, the surfaces were grit-blasted with 35 grit alumina using compressed air at 4 bars at a distance of 30 mm at an angle of 45. To remove abrasive residues and dust from the surfaces, the substrate samples were immersed in an ultrasonic ethanol solution for 15 min. After 1 min of grit blasting, the surface roughness (Ra) of the steel substrates was measured to be 3.2 µm by a tactile profilometer (Mitutoya Surftest SJ-210, Kawasaki, Japan). The cold spray method, on the other hand, creates in situ micro grit blasting with particles with a lower critical velocity impacting the surface of the substrate, leading to excellent bonding.

Figure 1 depicts the particle size distribution of spray powders determined with dynamic light scattering (Malvern Mastersizer 2000, Cambridge, UK) and gas-atomized Cu-10Sn bronze powders used as feedstock. XRD analysis (Bruker D2 Phaser, Berlin, Germany) of the spray powders was performed to scrutinize possible phase formations and alterations following deposition, as shown in Figure 2. In order to deposit the gas-atomized bronze powders, a portable Dymet 423 low-pressure cold spraying system (Dymet, Obninsk, Russia) was utilized. Table 1 shows the spray parameters used in this study. All specimens were coated using oil-free compressed dry and filtered air. To perform the microstructural observations, an optical microscope (OM; OLYMPUS/BX60M with an LC Micro v2.2 image analyzer, Tokyo, Japan) and a scanning electron microscope (SEM; Jeol JSM-6335F, Tokyo, Japan) equipped with an EDS unit were used to determine the constituents of the powders and the deposited layers. The hardness of the coatings was measured with micro Vickers hardness (HV) using a PC-controlled tester (Shimadzu HMV microhardness tester, Kyoto, Japan) at a load of 30 g for a dwell time of 15 s.

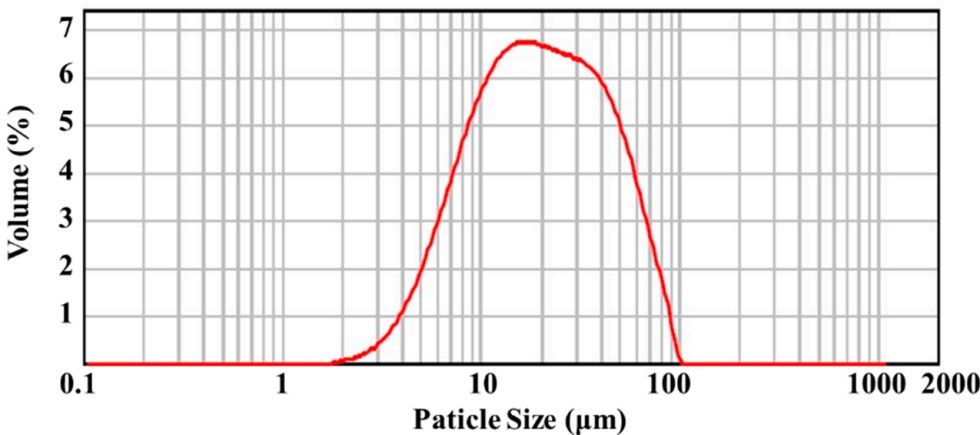

**Figure 1.** Particle size distribution of the gas-atomized Cu-10Sn powders used in the experiments.

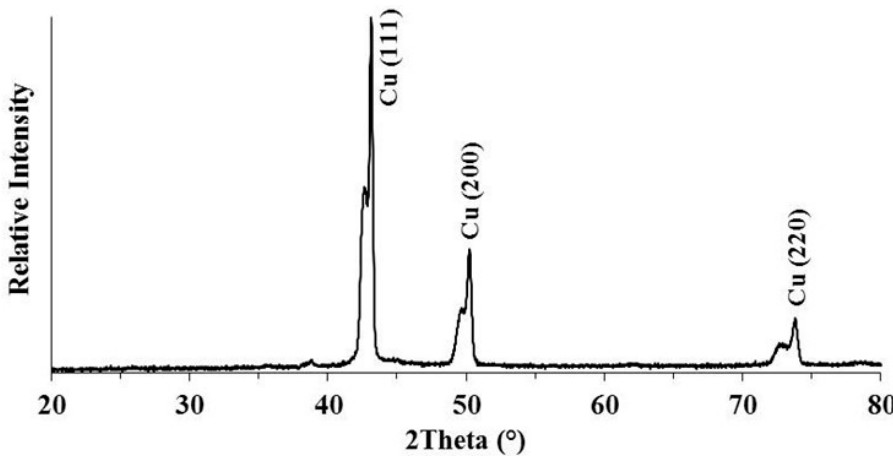

**Figure 2.** XRD spectrum of the Cu–10Sn powders.

**Table 1.** Cold spray parameters for Cu-10Sn coating deposition.

| Spray Parameters | Value |
| --- | --- |
| Gas pressure, bar | 8 |
| Gas temperature, °C | 400 |
| Powder feed rate, g·min$^{-1}$ | 105 |
| Spray distance, mm | 15 |
| Nozzle length, mm | 20 |
| Nozzle diameter, mm | 5 |
| Number of passes | 6 |

### 2.2. Wear and Corrosion Tests

For the friction and wear studies, a pin-on-disc tribometer (CSM, Neuchâtel, Switzerland) was used. The coatings were subjected to wear tests against an $Al_2O_3$ ball with a diameter of 5 mm under dry sliding conditions. The test speed remained constant (linear speed, 30 cm s$^{-1}$) while the normal load varied from 4 N to 8 N. The variation in friction coefficients was measured while increasing the sliding distance from 400 to 1000 m. Before the wear tests, the sprayed coatings were polished to an average roughness of Ra = 0.3 μm and rubbed on an alumina ball with a mirror-finished surface. In general, the surface roughness of typical bronze bearings produced by sintering was decreased to Ra = 0.5 μm by a fine boring process to a thickness of at least 200 microns. In contrast to sintered bearings, which required at least 300-micron layer removal, fine boring operations were performed precisely to reduce the thickness of the CS coatings at each level with satisfactory chip breaking.

The friction force was measured using a force transducer and constantly recorded in a computer. The friction coefficient was determined by dividing the friction force by the applied load. The wear rate was computed by dividing the worn volume by the applied force and sliding distance (mm$^3$/Nm$^{-1}$).

Electrochemical investigations of the coatings were performed with the potentiodynamic scanning (PDS) technique. All experiments were carried out with a computer-controlled potentiostat (PCI4/750, GAMRY) in 3.5 wt.% NaCl (alkaline) and 0.01 M $Na_2SO_4$ (acidic) solutions at room temperature. The alkaline and acidic environments had pH values of around 6.90 (+/− 0.005) and 5.75 (+/− 0.005), respectively. The electrochemical properties of the coatings were examined in aerated aqueous solutions.

Ag/AgCl (in saturated) and platinum (Pt) wire electrodes were used as a reference and auxiliary electrode, respectively. First, the specimens were immersed in the solution until reaching a steady open circuit potential (OCP) before starting the PDS tests. After equilibration, scanning was started at −0.3 V vs. Eocp, with a scanning rate of 1 mV/s. The exposed area of the test specimens was about 5 × 5 mm (±0.01 mm), and all of the

reported data have been normalized according to the surface area. The specimens were cleaned ultrasonically in ethanol after the corrosion tests to remove the corrosion products from the surfaces.

## 3. Results and Discussion

### 3.1. Microstructural Characterization

Figure 3 depicts the typical morphologies of the gas-atomized powders, which appeared as typically spherical. In comparison with irregularly shaped powders, spherical particles are preferable due to their superior flowability during feeding. Higher magnification of the as-received powders revealed that the grains had almost equiaxed appearances in all directions (see Figure 3b). The phase composition of the as-received powders is shown in Figure 2. As shown in the XRD spectrum, Cu peaks dominated the powders due to their significantly higher relative intensity peaks overlapping with Sn. It was obvious that no oxides or other phases existed in the powders in the received state. The spherical feedstock powders with a mean particle size of 20 μm were heated for 1 h at 80 °C before being fed into a powder feeder and sprayed onto the steel surfaces with the spray conditions listed in Table 1. A narrow particle size distribution with the same energetic accelerated particles resulted in good adhesion and homogenous coating with reduced porosity on the substrate. It is well-established that when the critical velocity is surpassed, accelerated powders at the nozzle exit have a tendency to pile up layer by layer on the substrate surface and form a coating. Furthermore, it was demonstrated that the critical velocity of the Cu particles was closely related to the oxygen content of the powder, which ranged from 300 to 610 m/s [18]. Figure 4 represents the typical cross-sectional microstructure of a cold-sprayed coating with a thickness of at least 450 μm. In contrast to conventional thermal spray methods, the observed extremely thick layers with no oxidation might be attributed to significant plastic deformation of particles without melting, void consolidation due to the hammering effect, and decreased oxidation and residual stresses created during deposition [10].

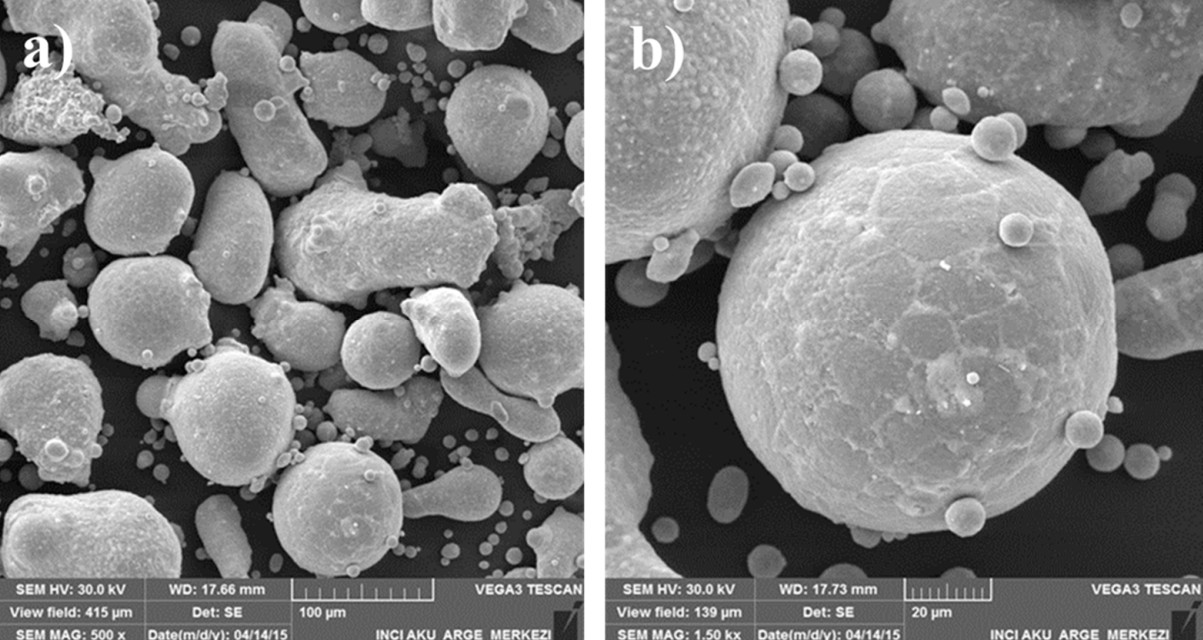

**Figure 3.** SEM micrographs showing Cu-10Sn powder shape after gas atomization process (**a**) with low and (**b**) high magnification.

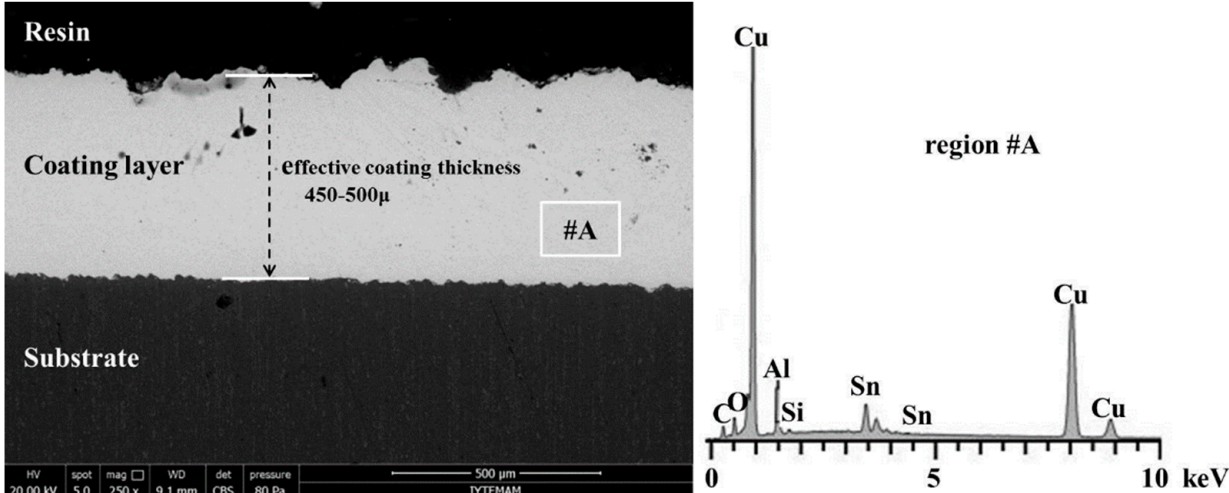

**Figure 4.** SEM image of the cross-sectional microstructure of the deposited coating layer and the EDS element concentrations at the coating cross section labeled #A.

It can be observed from the image that the Cu-10Sn powders adhered well to the substrate after impingement and layered up progressively, revealing a relatively dense structure on the surface of the substrate. This was accomplished by the conversion of kinetic energy to thermal energy during deformation, as well as peening of previously deposited layers, which resulted in decreased inter-splat voids and hence increased the coating density. Moreover, it was noticeable that the existence of microscopic pores in the coatings was inevitable.

The overall porosity predominantly distributed on top of the coating was measured as % 3.2 based on image analysis.

The first impinged particles exhibited intensive plastic deformation resulting in widespread flattening between the interfaces of the first deposited layers. However, owing to the strain-hardening effect, severely deformed initial layers lose deformability when impinged by incoming particles, resulting in increased porosity between the interlayers, though shot peening of accelerated particles with a lower critical velocity in-flight powder stream could further reduce porosity.

As a matter of fact, cold-sprayed bronze coatings were found to have greater porosity than pure Cu coatings. Additionally, this was ascribed to the lack of deformability of bronze powders due to their relatively high strength, as well as the distribution of pores, which was significantly linked to the Sn content of bronze powders [19]. Traditional sintered plain bearings with distinct surface porosity structures play an important role in providing an intrinsic self-lubrication effect. Furthermore, the tribological performance of a bearing was found to be significantly related to the pore size, network, distribution, and shape [20]. As a consequence, the functionality of cold spray coatings as plain-bearing candidates is dependent not only on their load-carrying capability but also on the reaction of the surface and the interior pore structure upon tribological interaction.

The EDS counts obtained from Cu K$\alpha$ and Sn K$\alpha$ showed that the deposited layer was mainly composed of Cu-Sn phase constituents, although some degree of $O_2$ capture occurred upon deposition, causing a decrease in deposition efficiency [18]. Figure 5 shows the phase composition carried out by means of X-ray diffraction of the deposited coating. According to the XRD analysis, the sprayed coating was primarily composed of Cu and Sn. Furthermore, no intermetallic compounds, metallic oxides, or new phases were evident in the Cu-10Sn coating XRD pattern. This could be attributed to a shorter in-flight particle dwell time in a relatively low-temperature gas stream (much below melting temperature) and the disruption of oxide films on particle surfaces during impacting, giving rise to fewer oxide forms inside the built-up layer.

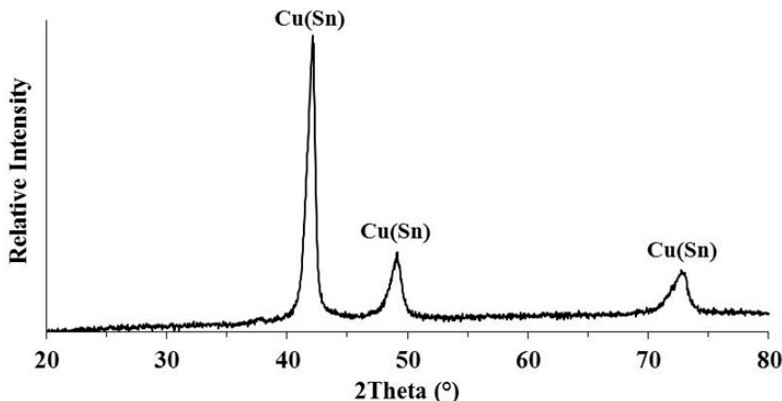

**Figure 5.** XRD patterns of the cold-sprayed Cu-10Sn coatings.

The hardness of the as-sprayed Cu-10Sn coatings was measured as 255 Hv$_{0.3}$. This value is much greater than that of conventional sintered Cu-10Sn bearing material (~120 Hv$_{0.3}$), which is strongly connected to the work-hardening effect caused by bounced-off bronze particles. Heat treatment of the as-sprayed coatings could enhance inter-splat bonding whilst reducing porosity. However, as compared to as-sprayed bronze coatings, this led to a higher wear rate and friction coefficient [19].

### 3.2. Wear and Corrosion Behavior

Figure 6 shows the variations in the friction coefficient (COF) of the CS coatings with increasing load and sliding distance. It can be seen that the friction coefficient of the sprayed coatings decreased from 0.77 (4 N, 400 cm) to 0.71 under 10 N over a 1000 m sliding distance as the applied load and run-in period increased. Furthermore, the coating evaluated at the maximum load (10 N) and longest sliding distance had a smooth COF characteristic, indicating that a steady-state condition was obtained. The relatively higher COF values of the coatings during sliding wear at the beginning were most likely owing to the inelastic interaction with the counter ball, which resulted in the fracture of the surface asperities and increased frictional contact area. The measured friction coefficient values were slightly greater than the values of coatings evaluated under dry sliding conditions sprayed with Cu or tin-based alloys reported in the literature [13,17,19].

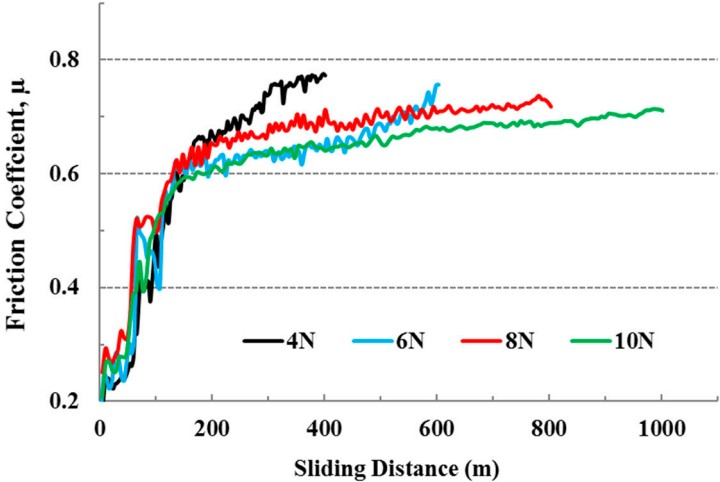

**Figure 6.** Variation in friction coefficient vs. sliding distance for as-sprayed Cu-10Sn under different loading conditions.

The surface qualities of deposited coatings as well as mechanical parameters, such as hardness, have a significant impact on tribological behavior. It was found that a higher

friction coefficient for cold-sprayed bronze coatings was correlated with an increase in Sn content in a Cu alloy [19]. In addition, heat-treated bronze coatings had greater friction coefficients than untreated bronze coatings, which was most likely due to low hardness accompanied by fragmented oxides or third-body particles during testing.

As stated in the introduction, additional coatings known as overlays could be employed to reduce the coefficient of the bearing surface. The several overlays deposited with different methods on the CuPb22Sn2 substrate indicated that tribological performance was based not only on reasonable sliding characteristics but also on adequate mechanical integrity and load-bearing capacity under optimal operating circumstances [1]. Incorporating solid lubricants, such as MoS2, h-BN, and graphite, into a polymer-based matrix could impart an intrinsic self-lubricating effect, allowing for reduced friction, particularly under poor lubrication conditions. The polymer matrix, on the other hand, needs to be sturdy enough to withstand increased heat or mechanical loading, otherwise the solid lubricant could stick out of the surface due to poor bonding and interface features [13]. Moreover, previous research demonstrated that h-BN particles could be incorporated into metallic powder by spray drying and a sintering technique and subsequently deposited onto a substrate in which a solid lubricant provided a repeatedly lubricating effect between mating surfaces during the running-in and therefore lowered the tendency of material flow at the surface.

Further examination of the as-sprayed coating microstructure, as shown in Figure 7, indicated that tiny size pores were evenly distributed throughout the microstructure, with relatively good interface bonding between flattened splat particles. Furthermore, the EDS mapping study revealed that Cu and Sn displayed widespread distribution and random coincidence across the microstructure, which mostly consists of a Cu-Sn-rich phase. EDS counts also revealed the formation of discrete oxide or new phases, while layering up was precluded. The variation in the wear rates of the as-sprayed coatings is illustrated in Figure 8. The wear rates of the as-sprayed coatings, such as the friction coefficient, decreased as the applied load increased. High hardness significantly lowered the coating wear rate once the interfacial bonding between flattened layers was improved, resulting in a reduction in the contact area, and thus the pores at the inter-splat boundary were minimal. The hard counter body served to induce milling action on the rough surface of the coating in the early stages of the wear test, and layer removal was accelerated through cutting and ploughing action. Nevertheless, as the load and sliding distance increased, Sn-rich ductile fragmented particles were pushed and smeared on the sliding ball surface, resulting in a substantial decrease in wear rate. Though post-heat treatment could be employed to reduce the adverse impacts of porosity and enhance the microstructure of the as-sprayed coating, diminished hardness increases the wear rate [19].

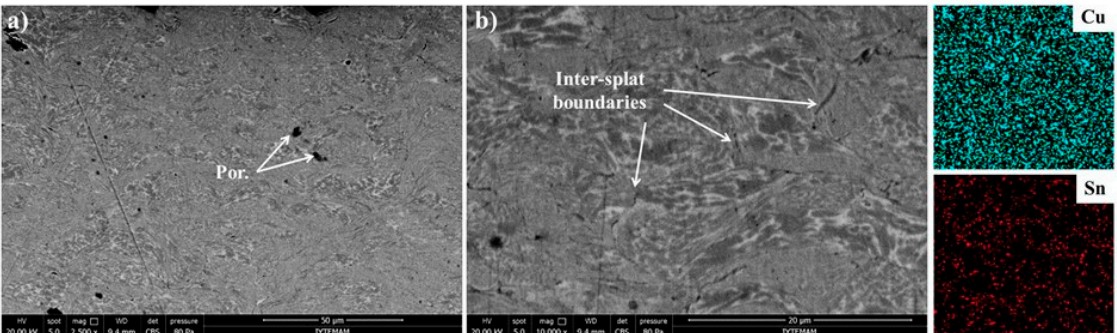

**Figure 7.** SEM cross-sectional microstructures at higher magnifications show porosity and flattened particle splat boundaries, as well as EDS maps with Cu and Sn element distribution.

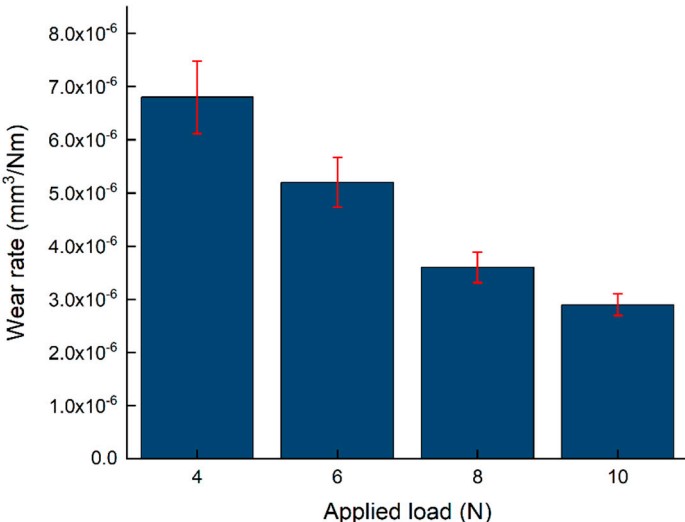

**Figure 8.** Specific wear rates of cold-sprayed Cu-10Sn coatings under varying applied loads.

Bearing performance during service, such as start and stop motions, might result in reduced bearing life due to material loss caused not only by excessive wear but also corrosion induced by interactions between oil degradation products and the bearing surface. The PDS responses of cold-sprayed samples obtained in acidic (0.01 M $H_2SO_4$) and alkaline (3.5% NaCl) environments are shown in Figure 9. To qualitatively evaluate the corrosion resistance of coatings, open circuit ($E_{ocp}$), corrosion ($E_{corr}$), pitting ($E_{pit}$), and corrosion current density ($I_{corr}$) values can be utilized. Therefore, the corrosion parameters computed from these curves are listed in Table 2. The Tafel technique, which is based on PDS experiments, was also used by the Gamry EchemAnalyst program. To compute $I_{corr}$, a slope in the cathodic branch was created at +100 mV from the intersection of the anodic and cathodic curves on the Tafel plot ($E_{corr}$). The corrosion current $I_{corr}$ was determined as the point where the slope line intersected the horizontal line derived from the $E_{corr}$.

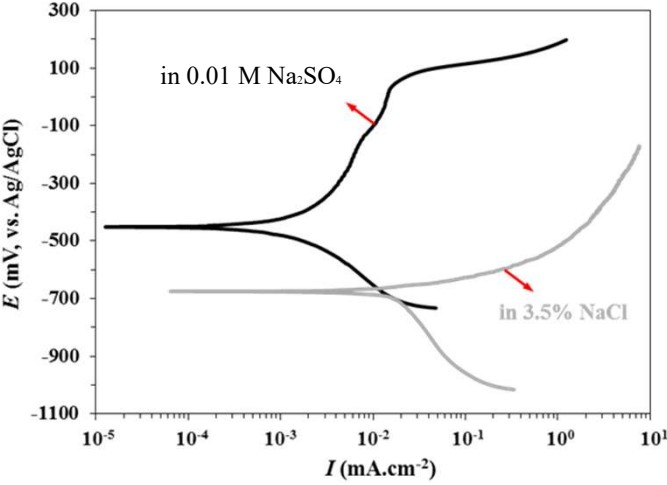

**Figure 9.** Potentiometric polarization curves of the Cu–10Sn cold-sprayed coatings in 3.5 wt % NaCl (alkaline) and 0.01 M $Na_2SO_4$ (acidic) electrolyte solutions at room temperature.

**Table 2.** Some important corrosion parameters calculated from PDS curves.

| Electrolyte | $E_{ocp}$ (mV) | $E_{corr}$ (mV) | $I_{corr}$ ($\mu A \cdot cm^{-2}$) | $E_{pit}$ (mV) |
|---|---|---|---|---|
| In 3.5% NaCl | −713 | −691 | 25.6 | - |
| In 0.01 M $Na_2SO_4$ | −432 | −450 | 3.7 | 43 |

Based on the PDS curves, $I_{corr}$ and $E_{corr}$ values of the coating in acidic media were lower than those of the coating in alkaline media. Furthermore, these curves show that no passive area evolved in an alkaline environment, as well as breaking potential in the 3.5% NaCl electrolyte. As a result, the dramatically decreased reactions were assumed to mask the corrosion processes in the solution. The higher $E_{corr}$ in acidic media indicated that the coating was more chemically stable and displayed a higher corrosion resistance. Furthermore, as compared to the alkaline environment condition, the $I_{corr}$ of the coating was dramatically lowered by a factor of seven orders of magnitude, from 25.6 μA·cm$^{-2}$ to 3.7 μA·cm$^{-2}$. The lower $I$corr implies that the coating has a lower corrosion rate as well as higher corrosion resistance [21]. The corroded surface morphologies of the coatings following PDS testing in alkaline and acidic solutions are shown in Figure 10. While the corrosion surfaces of the samples were comparable in both acidic and alkaline solutions, considerable differences could be observed. Compared with the surface morphology of the corroded coating in an alkaline solution, the surface of the coating exposed to an acidic electrolyte displayed more microscopic fractures and deeper grooves, indicating a preferential corrosion mechanism (Figure 10a,b).

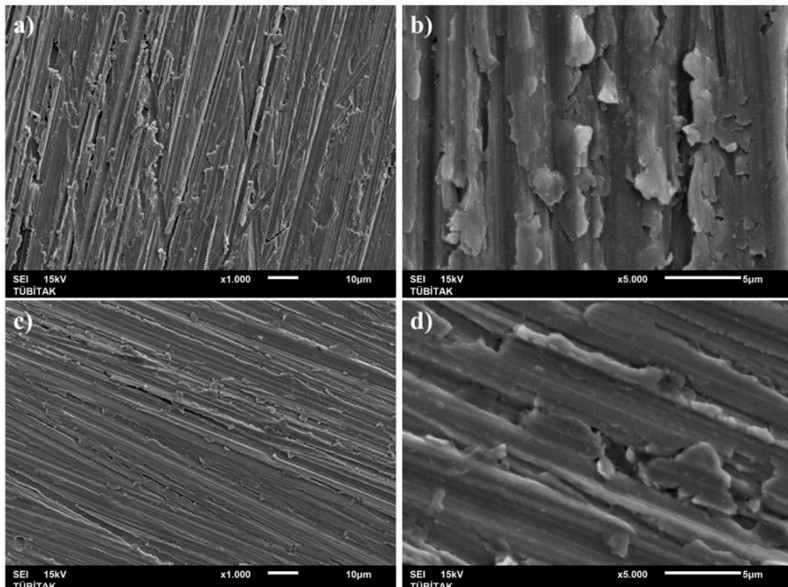

**Figure 10.** SEM images of the coating surfaces after corrosion tests (**a**,**b**) in Na$_2$SO$_4$ and (**c**,**d**) in NaCl solutions.

As a result, no measurable $E_{pit}$ value for the coating tested in the alkaline solution was recorded (Table 2); the corrosive medium permeated the interface between the electrochemical double layer and the coating surface. These observed micro peeling and channels may have created a direct path between the electrolyte and the substrate, resulting in rapid local galvanic attacks and pitting corrosion of the coating [22], resulting in no pitting potentials being shown in the PDS curve for the coating evaluated in the alkaline medium. It was recently revealed that the corrosion resistance of cold-sprayed Cu-based coatings is closely related to surface flaws, pore structure, and elemental contents. Furthermore, the relatively dense structure that was inversely proportional to thickness resulted in a decrease in the corrosion rate of the cold-sprayed Cu-based coatings [14].

Compared to the sintered typical plain-bearing structure shown in Figure 11, the cold spray process was shown to be more feasible and quicker in producing lead-free bronze-based plain bearings on steel back surfaces with tailored properties and excellent corrosion resistance. The porosities and thicknesses of CS metallic coatings have been found to determine their corrosion resistances after prolonged immersion in corrosive environments. Additionally, the pore size and distribution on the surface layer of a sprayed Cu coating,

through which corrosive liquids penetrate the coating, has a direct impact on the corrosion performance of the coating. It has been clearly demonstrated that cold-sprayed coating post-treatments, such as ball burnishing, shot blasting, and so on, may increase the integrity and bonding strength of coatings without cracking or splat removal, hence improving corrosion and wear performance [9,17]. A diamond smoothing post-treatment method will be employed in the future to examine the microstructural evolution at the surface in order to maximize the performance of the resultant bronze coating.

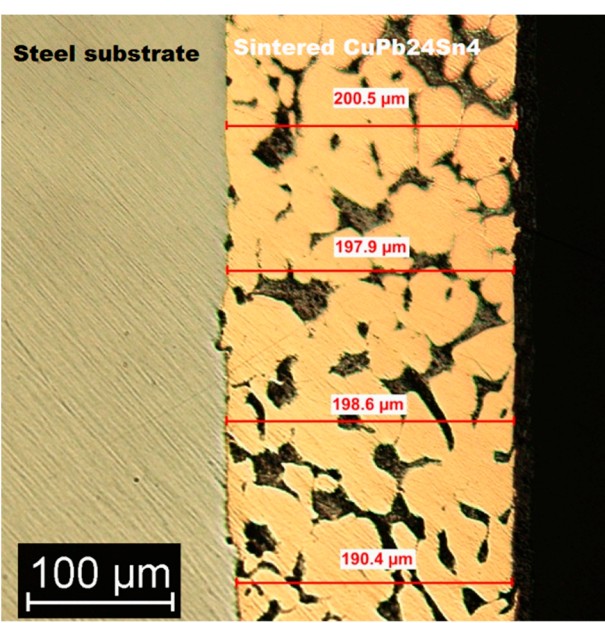

**Figure 11.** A typical cross-sectional view of the plain-bearing material sintered on a steel back, revealing Pb as black inside the microstructure.

Furthermore, in the absence of catastrophic failures, such as fatigue and seizure, worn-out bearing surfaces might be quickly refurbished using cold spray processes. As the cold spray additive manufacturing method expands, a wide range of industrial copper components with cylindrical shapes may be repaired or regenerated for real-time applications [10]. Despite the fact that it was technically challenging, machining the sprayed deposits was required to meet geometric, dimensional, and roughness tolerances [23]. The influence of surface roughness created by face turning with various indexable inserts on granulated surfaces, as well as dimensional control, will be evaluated, with a particular emphasis on chip formation and dimensional control. After this, the bearing material will be examined under service conditions in the near future after an overlay is applied to the finished surface of the CS bronze coating.

### 4. Conclusions

The gas-atomized Cu-10Sn particles were sprayed onto the steel substrate using a low-pressure cold spray technique. The mechanical properties of the sprayed coatings, such as friction, wear, and hardness, were examined. Potentiodynamic scanning (PDS) was used to evaluate the corrosion susceptibilities of the composite coatings in acidic (0.01 M $Na_2SO_4$) and alkaline (3.5% NaCl) environments. The main results of this study can be summarized as follows:

1. Low-pressure cold-sprayed gas-atomized Cu-10Sn powders were effectively deposited onto the steel substrate with at least 450 μm effective thickness, displaying tiny-sized and well-distributed pores with a progressive modest increase through the top layer due to the hammering action of incoming particles during the cold spray deposition process. After deposition, the coating exhibited no detectable oxide and the creation of a new phase. Fine boring operations were carried out successfully to

lower the thickness of the CS coating at each level, with good chip breaking to the surface roughness (Ra = 0.5 micron) and no observable surface cracking.

2.  The hardness of the as-sprayed Cu-10Sn coatings (255 $Hv_{0.3}$) was substantially higher than that of the conventional sintered Cu-10Sn bearing material (120 $Hv_{0.3}$), which was directly related to the hammering action of the colliding particles with high kinetic energy.

3.  A steady-state condition was reached for the coatings evaluated in the ball-on-disc dry sliding test with increasing load and sliding distance, resulting in a 0.71 friction coefficient under 10 N over a 1000 m sliding distance. As the applied load increased from 4 N to 10 N, the wear rates of the as-sprayed coatings decreased by a factor of three, which was attributed to Sn-rich ductile particles smeared on mating surfaces, leading to a significant drop in wear rate.

4.  The PDS findings revealed that the corrosion resistance of the Cu-10Sn coating layer was greater in an acidic environment than in an alkaline environment. In addition, due to the strong corrosion reaction in an alkaline solution, the coated layer showed no passivation or pitting onset.

**Author Contributions:** Conceptualization, I.O. and B.B.; methodology, I.O. and B.B.; validation, I.O., B.B. and T.G.; formal analysis, I.O.; investigation, I.O. and B.B.; resources, B.B. and T.L.; data curation, I.O.; writing—original draft preparation, I.O.; writing—review and editing, I.O., T.G. and T.L.; visualization, I.O.; supervision, T.L. All authors have read and agreed to the published version of the manuscript.

**Funding:** Alexander von Humbolt Foundation 10. Round PSI.

**Data Availability Statement:** Not applicable.

**Acknowledgments:** This work was supported by the Philipp Schwartz Initiative of the Alexander von Humboldt Foundation.

**Conflicts of Interest:** The authors declare that there are no conflicts of interest in this work.

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
