# Peer review of "Wear and Corrosion Behavior of Cold-Sprayed Cu-10Sn Coatings"

_crystals, doi:10.3390/cryst13030523_

Round 1

Reviewer 1 Report

In the paper "Wear and Corrosion Behaviour of Cold Sprayed Cu-10Sn Coating", the mechanical properties of sprayed coatings such as friction, wear, and hardness were investigated, and their corrosion resistance was evaluated. Although the paper is fairly well-written, it contains certain errors and shortcomings.

Changes which must be made before publication:

   The introduction lacks information on the literature regarding the corrosion resistance of the investigated coatings.

   The authors state that corrosion tests were performed in both acidic and alkaline environments, but do not provide information on the pH of the corrosive solutions used. For instance, it is not clear what was used to acidify Na2SO4.

   There is also no information on whether the corrosive solution was aerated or in contact with air, which can strongly affect the corrosion resistance of the tested material.

   Additionally, there is no information on how the corrosion rate (ikor) was determined.

   On page 8, line 244, the authors write: "The lower Ecorr in acidic media indicated that the coating..." - is this really the case? It should rather be higher.

On page 8, line 250, the authors write: "In comparison to the surface morphology of corroded coating in an acidic solution, the surface of coating exposed to Na2SO4  " - it should rather be "In comparison to the surface morphology of corroded coating in an alkaline solution.

Author Response

Dear Reviewer,

Please see the attached for the Author's Comment to the Review Report.

Thank you so much for your cooperation and contributions. 

Sincerely Yours

Reviewer 2 Report

Dear authors,

line 270: please consider 'comparison' to replace 'compared'

lines 141-171; 175-223, 232-262.  Please consider a few paragraph breaks, as needed, when a topic is changed.

Author Response

(The authors gave the same response as above.)

Reviewer 3 Report

Wear and Corrosion Behaviour of Cold Sprayed Cu-10Sn Coating

·         The steel substrate was coated with Cu-10Sn particles that had been atomized using gas and sprayed onto the surface using a low-pressure cold spray technique. The resulting coatings were analyzed to determine their mechanical properties, including friction, wear, and hardness. To assess their corrosion susceptibility, potentiodynamic scanning (PDS) was used in both acidic (0.01 M Na2SO4) and alkaline (3.5% NaCl) environments.

 GENERAL IDEA IS ok, BUT PAPER SI SHORT.

·         I do not like the definition of bearing. My recommendation. Eliminate general statements and the first two lines of introduction.

·         Which typo of bearing: friction ones, hydrodynamics…please define it better. H.Urreta works are worth reading see: Seals based on magnetic fluids for high precision spindles of machine tools, International Journal of Precision Engineering and Manufacturing 19 (4), 495-503 and the extension in Journal of intelligent material systems and structures 30 (15), 2257-2271 These are key for bearings. Yours are not using the same magnetorheolgy approaches but definition are here Ok.

·         On the other hand some works are missed when sprays surfaces are referred. State of the art missed several approaches. Cold sprays and other, even thermal sprays, usually are concerned with bonding, when other operations are applied for finishing. I read a work Turning of thick thermal spray coatings, Journal of thermal spray technology 10 (2), 249-254 and it opened several research lines. Please include some consideration in introduction of at the final explanations.

·         SEM micrographs showing Cu-10Sn powder shape after gas atomization process: are they similar in size to other additive methods?

·         n, flat form steel substrates 40´40´8 mm in dimensions were 67 cut from conventional steel shell (AISI 1010) by using an abrasive cutting machine and 68 grit blasted with corundum abrasive particles. This is poorly explained: which final roughness duid you get? Please define better the surface state. Pretreatments by ball burnings, shot peening or laser shock peening are available today. The works you missed define for instance that coating bonding depends a lot on it.

Regarding the bonding of the coating, it appears to be highly dependent on various factors that may influence its adhesion to the surface. Therefore, it may be beneficial to have a more in-depth discussion on the final application to better understand the specific requirements and potential challenges that may affect the coating's performance. Additionally, while your definition of a bearing is noted, it may be helpful to review the current state of the art to ensure a comprehensive understanding of the topic. Furthermore, it is suggested that an extended version be considered that incorporates suggestions to improve the overall quality and effectiveness of the discussion. Thank you for your consideration and willingness to collaborate towards achieving the best outcome.

Are the EDS graphs meaningful for the reading?

With all due respect, may I suggest that the current state of the art section could benefit from further elaboration and clarification? Specifically, it may be helpful to provide more detail on the existing research and developments in the field, including the most recent findings and advancements

Author Response

(The authors gave the same response as above.)

Round 2

Reviewer 3 Report

Paper is OK, thanks